# 3α,7-Dihydroxy-14(13→12)*abeo*-5β,12α(H),13β(H)-cholan-24-oic Acids Display Neuroprotective Properties in Common Forms of Parkinson’s Disease

**DOI:** 10.3390/biom13010076

**Published:** 2022-12-30

**Authors:** Andreas Luxenburger, Hannah Clemmens, Christopher Hastings, Lawrence D. Harris, Elizabeth M. Ure, Scott A. Cameron, Jan Aasly, Oliver Bandmann, Alex Weymouth-Wilson, Richard H. Furneaux, Heather Mortiboys

**Affiliations:** 1Ferrier Research Institute, Victoria University of Wellington, Lower Hutt 5040, New Zealand; 2Sheffield Institute for Translational Neuroscience (SITraN), Department of Neuroscience, University of Sheffield, 385a Glossop Road, Sheffield S10 2HQ, UK; 3Department of Neurology, St. Olav’s Hospital, 7006 Trondheim, Norway; 4ICE Pharma, 68 Weld Street, RD2, Palmerston North 4472, New Zealand

**Keywords:** bile acids, C-*nor*-D-*homo* bile acids, rearrangement, drug discovery, neurodegenerative diseases, Parkinson’s Disease, UDCA, LRRK2

## Abstract

Parkinson’s Disease is the most common neurodegenerative movement disorder globally, with prevalence increasing. There is an urgent need for new therapeutics which are disease-modifying rather than symptomatic. Mitochondrial dysfunction is a well-documented mechanism in both sporadic and familial Parkinson’s Disease. Furthermore, ursodeoxycholic acid (UDCA) has been identified as a bile acid which leads to increased mitochondrial function in multiple in vitro and in vivo models of Parkinson’s Disease. Here, we describe the synthesis of novel C-*nor*-D-*homo* bile acid derivatives and the 12-hydroxy-methylated derivative of lagocholic acid (**7**) and their biological evaluation in fibroblasts from patients with either sporadic or LRRK2 mutant Parkinson’s Disease. These compounds boost mitochondrial function to a similar level or above that of UDCA in many assays; notable, however, is their ability to boost mitochondrial function to a higher level and at lower concentrations than UDCA specifically in the fibroblasts from LRRK2 patients. Our study indicates that novel bile acid chemistry could lead to the development of more efficacious bile acids which increase mitochondrial function and ultimately cellular health at lower concentrations proving attractive potential novel therapeutics for Parkinson’s Disease.

## 1. Introduction

Parkinson’s Disease (PD) is the most common neurodegenerative movement disorder globally, with its prevalence rising. Currently, there are approximately 10 million people worldwide living with PD and this is estimated to increase to 13.5 million by 2030 [1]. Most cases of PD are sporadic in nature with no known origin. However, a subset of PD patients has known genetic causes, most commonly in the gene encoding the leucine-rich repeat kinase 2 (LRRK2); furthermore, we are understanding more about the genetic risk factors associated with sporadic PD (sPD) after the publication of the large genome-wide association studies (GWAS) [2]. Symptomatic treatment has been available for many years for PD to compensate for the decline in dopamine release as the dopaminergic neurons are progressively lost in the course of the disease, but the quest for new disease-modifying therapies is urgent. A recent review highlighted the potential new treatments, both disease-modifying and symptomatic, which are currently in clinical trials [3]. Although many potential therapeutics are now in clinical trials, more work is urgently needed to find new potential disease-modifying treatments.

Over the last decade, increasing evidence has established a compelling link between mitochondrial abnormalities and PD. The first clue in this matter was uncovered when a toxic contaminant of a recreational drug, 1-methyl-4-phenyl-1,2,3,6-tetrahydropyridine (MPTP), induced Parkinsonism. This compound was later identified as a mitochondrial complex I inhibitor [4]. Since then, mitochondrial damage has been found in tissue from both sporadic [5,6,7,8,9] and familial PD patients [7,10,11,12,13]. The latter frequently harbor mutations in most genes that are associated with PD, including PRKN, PINK1 and LRRK2. Today, mitochondrial dysfunction is recognized as one of the major cellular mechanisms associated with both familial and sporadic PD and many active pre-clinical and clinical research studies are aimed to leverage this mechanism (in addition to lysosomal abnormalities) for the development of new drugs [3]. Because mitochondrial dysfunction presents in peripheral tissues from PD patients, these can be used to assess compounds for their ability to correct mitochondrial deficits. Previously, we have undertaken a screen of 2000 compounds in fibroblasts derived from patients with PRKN mutations and identified ursodeoxycholic acid (UDCA) as a compound which was capable of recovering some of the mitochondrial deficits in these fibroblasts [12]. Moreover, UDCA was found to improve the mitochondrial function in fibroblasts from patients with LRRK2G2019S mutation and a subset of sporadic PD patients [7,12]. As a result of its potential as a PD-modifying drug, UDCA has been advanced into clinical trials for PD [3,14]. These outcomes also underscore that compound screening in primary patient fibroblasts has translational potential to identify candidates which can be of benefit in a clinical setting. 

Recently, we reported the synthesis of the 12β-methyl-18-*nor*-chenodexoycholic acid (**1**) and its 17-*epi*-counterpart (**2**) from cholic acid (Figure 1) [15]. In the course of that work, we also isolated the 14(13→12)-*abeo*-bile acid **3** as a minor side product. Its ring system, containing a five-membered C-ring and a six-membered D-ring, is unique in that bile acids of this type are not known to the best of our knowledge. However, naturally occurring steroidal alkaloids of the veratrum family exhibit precisely this C-*nor*-D-*homo* framework [16,17]. Best known among these alkaloids is cyclopamine (**5**) which was isolated from the corn lily *Veratrum californicum* and found to be the underlying cause of cyclopia, a rare congenital disorder that leads to the offspring being born with only a single eye in the center of their face. Subsequent research into the mechanism of action of cyclopamine (**5**) established that it antagonizes the seven-pass transmembrane protein Smoothened (Smo; IC_50_ = 5000 nM) in the hedgehog (Hh) signaling pathway [18,19]. Cyclopamine and analogues also hold promise for the treatment of cancer [19,20,21,22].

In this paper, we describe the synthesis of **3**, its taurine conjugate (**24**) as well as the 7β-configured analogue **4**. These compounds were part of a wider library of 12β-methyl-18-*nor*-bile acids that were screened for activity in various bile acid-relevant assays [15,23,24]. Screening of these compounds in fibroblasts from patients with Parkinson’s Disease revealed compound **4** and the taurine conjugate of **3** as potential candidates for further assessment. In addition, the 12-hydroxy-methylated derivative of lagocholic acid **7**, emerged as a compound of interest from this screen. Consequently, these compounds were examined further for their ability to restore mitochondrial function in comparison to UDCA. Hence, we report their effects on modulating cellular ATP levels as well as the mitochondrial membrane potential as key parameters in both sporadic PD patient fibroblasts and LRRK2G2019S patient fibroblasts. Among the compounds tested, **4** and **7** showed the greatest promise, displaying an increase in both ATP and MMP levels that were close to or better than those exerted by UDCA. Moreover, these effects were produced at lower concentrations than UDCA, indicating that they are more efficacious. This was particularly true for the MMP assay, with the cellular ATP effect less robust in the sPD fibroblasts.

## 2. Materials and Methods

### 2.1. Chemistry 

#### 2.1.1. General Experimental Procedures

Melting points were determined by differential scanning calorimetry (DSC). Proton (^1^H) and carbon (^13^C) NMR spectra were recorded on Bruker Avance (III)-500. Chemical shifts are reported in ppm relative to Me_4_Si (TMS, d 0), or residual solvent peaks as an internal standard set to δ 7.26 and 77.00 (CDCl_3_), or δ 3.34 and 49.05 (CD_3_OD). NMR data are reported as follows: chemical shift in ppm, multiplicity (s = singlet, d = doublet, t = triplet, sp = septet, dd = doublet of doublets, ddd = doublet of doublets of doublets, td = triplet of doublets, dt = doublet of triplets, m = multiplet), coupling constant in Hz, integration.

Electrospray ionization (ESI) mass spectrometry (MS) experiments were performed on a quadrupole time-of-flight (QTOF) Premier mass spectrometer (Micromass, UK) under normal conditions. Sodium formate solution was used as a calibrant for high-resolution mass spectra (HRMS) measurements. Specific optical rotations were acquired on a Rudolph Autopol^®^ IV Automatic polarimeter at ambient temperature (20 °C), unless otherwise stated, λ = 589 nm and concentration (g/100 mL) in the solvent indicated, using a cell of path length 100 mm

All reactions were monitored by thin layer chromatography (TLC) using 0.2 µm silica gel (Merck Kieselgel 60 F_254_) precoated aluminium plates, using UV light, ammonium molybdate or potassium permanganate staining solution to visualize. Flash column chromatography was performed on Davisil^®^ silica gel (60, particle size 0.040–0.063 mm), or using Reveleris^®^ silica or C-18 reversed phase flash cartridges on a Grace Reveleris^®^ automated flash system with continuous gradient facility. Solvents for reactions and chromatography were analytical grade and were used as supplied unless otherwise stated. Unless otherwise stated petroleum ether refers to the fraction boiling at 60–80 °C.

The purity of all relevant compounds was evaluated by high-performance liquid chromatography (HPLC) or liquid chromatography-mass spectrometry (LCMS) and determined to be ≥95%. The analytical LCMS analyses were carried out on an Agilent 1260 Infinity II Series LC System with an Agilent 6120B Single Quadrupole LC/MS (ESI), equipped with an Agilent 1100 Multi Wavelength Detector (WD) and an Agilent Infinity II 1290 Evaporative Light Scattering Detector (ELSD). High-performance liquid chromatography (HPLC) analyses were performed on an Agilent 1100 (Quaternary pump) HPLC system with a diode array detector (200–400 nm) and refractometric index (RI) detector, employing columns as indicated in the Appendix A. Injection volumes were typically 30 μL (1–2 mg·mL^−1^) and data were processed with Agilent Cerity System software.

#### 2.1.2. Syntheses

Methyl 3α,7α-diacetyloxy-12-oxo-5β-cholan-24-oate (**17**) [25,26,27]. To a solution of **16** (6.76 g, 13.3 mmol) and sodium acetate (13.6 g, 166 mmol) in methanol (100 mL) was added bromine (6.2 mL, 121 mmol) dropwise over 3 h at room temperature. After being stirred overnight the reaction was quenched with a saturated solution of sodium thiosulphate and diluted further with water. The aqueous was extracted with ethyl acetate (3×) and the combined organic fractions were washed with brine, dried over MgSO_4_ and concentrated. The crude reaction product was purified by automated column chromatography (silica gel, ethyl acetate/ petroleum ether 0–30%) to afford 6.42 g (95%) of **17** as a colorless solid. The spectroscopic data of **17** were consistent with those reported for the same compound in [28].

Methyl 3α,7α-diacetyloxy-12β-hydroxy-5β-cholan-24-oate (**18**). 

Method **A** [25,26]: To a solution of **17** (1.00 g, 1.98 mmol) in dry dichloromethane (50 mL) was added borane *tert*-butylamine complex (463 mg, 5.32 mmol) in one portion at room temperature. After being stirred for 2 h, the reaction was carefully quenched with 1 M aqueous hydrochloric acid solution and extracted with ethyl acetate (3×). The organic layers were combined, washed with water (1×) and brine, dried over MgSO_4_ and concentrated. The crude product was purified by automated column chromatography (silica gel, ethyl acetate/petroleum ether 10–100%) to yield 883 mg (88%) of a 1:2 mixture of **16** and **18** (as determined by HPLC analysis) as a colorless foam. 

Method **B**: To a solution of **17** (1.00 g, 1.98 mmol) and cerium trichloride heptahydrate (929 mg, 2.49 mmol) in a 2:1 mixture of methanol (36 mL) and tetrahydrofuran (18 mL) was added portionwise sodium borohydride (137 mg, 3.62 mmol) at 0 °C. After being stirred for 2 h the reaction was quenched with 1 M aqueous hydrochloric acid solution, diluted with water and extracted with ethyl acetate (3×). The organic layers were combined, washed with brine, dried over MgSO_4_ and concentrated. The crude product was purified by automated column chromatography (silica gel, ethyl acetate/petroleum ether 10–100%) to yield 746 mg (74%) of a 1:1.4 mixture of **16** and **18** (as determined by HPLC analysis) as a colorless foam.

The desired 12β-hydroxy methyl ester **18** was separated by repeated automated column chromatography [silica gel (25 µm), ethyl acetate/petroleum ether 10–80%] of portions of up to 800 mg using a 40 g high-performance (HP) cartridge to give a colorless solid. R*_f_* (ethyl acetate/petroleum ether 3:7) = 0.26 (**17**, SM), 0.18 (**18**), 0.15 (**16**); M.p. 182–183 °C (DSC); ${\left[\alpha \right]}_{D}^{20}$ = +28.5 (*c* 0.75, CHCl_3_); ^1^H NMR (500 MHz, CDCl_3_) δ 4.90 (dd, *J* = 5.7, 3.0 Hz, 1H), 4.59 (tt, *J* = 11.3, 4.5 Hz, 1H), 3.66 (s, 3H), 3.49–3.42 (m, 1H), 2.38 (ddd, *J* = 15.8, 9.1, 5.2 Hz, 1H), 2.27 (ddd, *J* = 15.9, 8.6, 7.4 Hz, 1H), 2.04 (s, 3H), 2.04–1.90 (overlapping signals: 2.03, s, 3H and m, 4H), 1.90–1.82 (m, 2H), 1.78–1.65 (m, 3H), 1.64–1.56 (m, 2H), 1.56–1.38 (m, 7H), 1.33–1.15 (m, 4H), 1.11 (td, *J* = 14.4, 3.5 Hz, 1H), 1.01 (d, *J* = 6.8 Hz, 3H), 0.94 (s, 3H), 0.73 (s, 3H); ^13^C NMR (125 MHz, CDCl_3_) δ 174.69, 170.57, 170.26, 79.01, 73.96, 70.79, 57.24, 51.49, 48.60, 47.79, 40.70, 36.67, 34.87, 34.74, 34.62, 33.24, 32.63, 32.15, 31.29, 30.94, 29.29, 26.76, 23.66, 22.97, 22.52, 21.53, 21.43, 21.04, 7.67; HRMS (ESI) m/z calcd for C_29_H_46_O_7_Na^+^ 529.3136, found 529.3138.

Methyl 3α,7α-diacetyloxy-12β-methoxy-5β-cholan-24-oate (**19**). To a solution of **18** (200 mg, 0.395 mmol) in dry *N*,*N*-dimethylformamide (DMF; 6 mL) was added sodium hydride (60% in mineral oil; 18.9 mg, 0.474 mmol) at 0 °C. After being stirred for 30 min iodomethane (0.16 mL, 2.57 mmol) was introduced before the reaction was allowed to warm up to room temperature overnight. Then the reaction was quenched with water and extracted with ethyl acetate (3×). The organic fractions were combined, washed with bine, dried over MgSO_4_ and concentrated. The crude reaction product was purified by automated flash column chromatography (silica gel, ethyl acetate/petroleum ether 0–20%) to afford 167 mg (81%) of **19**. ${\left[\alpha \right]}_{D}^{20}$ = +14.7 (*c* 1.36, CHCl_3_); ^1^H NMR (500 MHz, CDCl_3_) δ 4.89 (dd, *J* = 5.8, 3.1 Hz, 1H), 4.60 (tt, *J* = 11.3, 4.5 Hz, 1H), 3.66 (s, 3H), 3.31 (s, 3H), 2.89 (dd, 10.8, 4.5 Hz, 1H), 2.37 (ddd, *J* = 15.4, 10.1, 5.2 Hz, 1H), 2.25 (ddd, *J* = 15.4, 9.9, 6.4 Hz, 1H), 2.04 (s, 3H), 2.03–1.97 (overlapping signals: m, 1H and 2.03, s, 3H), 1.94 (ddd, *J* = 15.6, 5.6, 3.7 Hz, 1H), 1.92–1.80 (m, 5H), 1.80–1.67 (m, 2H), 1.64–1.56 (m, 2H), 1.55–1.36 (m, 6H), 1.35–1.07 (m, 5H), 0.98 (d, *J* = 6.9 Hz, 3H), 0.94 (s, 3H), 0.68 (s, 3H); ^13^C NMR (125 MHz, CDCl_3_) δ 174.69, 170.54, 170.22, 88.70, 73.99, 70.95, 57.38, 56.63, 51.41, 48.75, 47.46, 40.75, 36.97, 34.93, 34.86, 34.63, 32.98, 32.61, 32.45, 31.30, 29.91, 26.84, 25.34, 24.13, 22.94, 22.53, 21.51, 21.42, 20.66, 8.33; HRMS (ESI) m/z calcd for C_30_H_48_O_7_Na^+^ 543.3292, found 543.3295.

3α,7α-Dihydroxy-12β-methoxy-5β-cholan-24-oic acid (**7**). To a solution of **19** (157 mg, 0.302 mmol) in methanol (10 mL) was added an 8 M aqueous solution of potassium hydroxide (1.1 mL) and the resulting mixture was heated at 80 °C overnight. After complete deprotection (TLC analysis) the reaction was cooled to room temperature and the methanol was evaporated off under reduced pressure. The residue was diluted with water and subsequently acidified to pH 1 by introducing dropwise a 1 M aqueous solution of hydrochloric acid. This was then extracted with ethyl acetate (3×) and the combined organic fractions were washed with brine, dried over MgSO_4_ and concentrated. The crude reaction product was finally purified by automated column chromatography [silica gel, acetone (+1% acetic acid)/dichloromethane (+1% acetic acid) 0–30%] to yield 95.1 mg (75%) of **7** as a colorless amorphous solid. ${\left[\alpha \right]}_{D}^{20}$ = +7.6 (*c* 0.49, EtOH); ${\left[\alpha \right]}_{D}^{20}$ = +9.0 (*c* 0.52, MeOH); ^1^H NMR (500 MHz, CD_3_OD) δ 3.82–3.79 (m, 1H), 3.38 (tt, *J* = 11.1, 4.4 Hz, 1H), 3.31 (s, 3H), 2.92 (dd, *J* = 10.9, 4.4 Hz, 1H), 2.34 (ddd, *J* = 15.3, 10.0, 5.3 Hz, 1H), 2.29–2.16 (m, 2H), 1.99–1.82 (m, 6H), 1.82–1.73 (m, 2H), 1.68–1.61 (m, 2H), 1.59–1.46 (m, 3H), 1.46–1.33 (m, 4H), 1.33–1.18 (m, 2H), 1.16–0.99 (overlapping signals: m, 2H and 1.01, d, *J* = 6.9 Hz, 3H), 0.93 (s, 3H), 0.70 (s, 3H); ^13^C NMR (125 MHz, CD_3_OD) δ 178.31, 90.68, 72.81, 68.82, 58.81, 56.93, 50.06, 48.60, 43.05, 40.53, 39.81, 36.60, 36.41, 36.00, 34.19, 33.56, 33.15, 31.42, 31.32, 26.58, 25.27, 24.06, 23.31, 21.26, 8.98; HRMS (ESI) m/z calcd for C_25_H_42_O_5_Na^+^ 445.2924, found 445.2935.

*iso*-Propyl 7α-acetyloxy-3α,12β-dihydroxy-5β-cholan-24-oate (**21**) and *iso*-propyl 7α-acetyloxy-3α,12α-dihydroxy-5β-cholan-24-oate (**20**). To a mixture of regular aluminium foil (1.53 g, 56.7 mmol) in dry *iso*-propanol (25 mL) was added mercury(II) chloride (62.7 mg, 0.231 mmol) and the mixture was heated under reflux until the aluminium was dissolved. Subsequently, **17** (1.00 g, 1.98 mmol) was added in one portion and the reaction was heated for another 3.5 h at reflux. After being cooled to room temperature, the reaction was quenched with water and then acidified to pH 1 by introducing dropwise a 1 M aqueous solution of sulfuric acid. The resulting mixture was extracted with ethyl acetate (3×) and the combined organic phases were washed with water (1×) and brine, dried over MgSO_4_ and concentrated. The crude product mixture was purified by automated column chromatography (silica, acetone/dichloromethane 2–80%) to afford 500 mg (51%) of **21** and 286 mg (29%) of **20**. R*_f_* (acetone/dichloromethane 3:7) = 0.50 (**21**, major) and 0.26 (**20**); 

*iso-Propyl 7α-acetyloxy-3α,12β-dihydroxy-5β-cholan-24-oate* (**21**). ^1^H NMR (500 MHz, CDCl_3_) δ 4.99 (sp, *J* = 6.3 Hz, 1H), 4.90 (dd, *J* = 5.7, 3.0 Hz, 1H), 3.50 (tt, *J* = 10.9, 4.5 Hz, 1H), 3.45 (dd, *J* = 11.1, 4.8, Hz, 1H), 2.34 (ddd, *J* = 15.6, 9.1, 5.4 Hz, 1H), 2.22 (ddd, *J* = 15.8, 8.4, 7.5 Hz, 1H), 2.05 (s, 3H), 1.98–1.81 (m, 6H), 1.78–1.39 (m, 12H), 1.39–1.31 (m, 1H), 1.31–1.16 overlapping signals (m, 4H and 1.22, d, *J* = 6.3 Hz, 6H), 1.09–0.99 overlapping signals (1.05, td, *J* = 14.3, 3.4 Hz, 1H and 1.02, d, *J* = 6.8 Hz, 3H), 0.93 (s, 3H), 0.73 (s, 3H); ^13^C NMR (125 MHz, CDCl_3_) δ 173.82, 170.45, 78.95, 71.57, 70.90, 67.44, 57.19, 48.61, 47.77, 40.84, 38.84, 36.68, 35.14, 34.67, 33.23, 32.67 (2C), 31.38, 30.91, 30.55, 29.31, 23.80, 22.94, 22.54, 21.84, 21.82, 21.54, 21.00, 7.66; HRMS (ESI) m/z calcd for C_29_H_48_O_6_Na^+^ 515.3343, found 515.3351.

*iso-Propyl 7α-acetyloxy-3α,12α-dihydroxy-5β-cholan-24-oate* (**20**). ^1^H NMR (500 MHz, CDCl_3_) δ 5.00 (sp, *J* = 6.3 Hz, 1H), 4.91–4.86 (m, 1H), 4.01–3.97 (m, 1H), 3.49 (tt, *J* = 11.0, 4.5 Hz, 1H), 2.32 (ddd, *J* = 15.2, 9.8, 5.3 Hz, 1H), 2.25–2.15 (m, 2H), 2.06 (s, 3H), 2.04–1.75 (m, 6H), 1.75–1.50 (m, 8H), 1.48–1.25 (m, 7H), 1.22 (d, *J* = 6.3 Hz, 6H), 1.13–0.95 overlapping signals (m, 2H and 0.98, d, *J* = 6.5 Hz, 3H), 0.91 (s, 3H), 0.68 (s, 3H); ^13^C NMR (125 MHz, CDCl_3_) δ 173.67, 170.65, 72.72, 71.74, 71.00, 67.37, 47.20, 46.57, 42.07, 41.12, 38.86, 38.14, 35.08, 34.95, 34.32, 31.62, 31.41, 30.88, 30.55, 28.55, 28.15, 27.27, 22.96, 22.55, 21.83 (2C), 21.63, 17.34, 12.48; HRMS (ESI) m/z calcd for C_29_H_48_O_6_Na^+^ 515.3343, found 515.3350.

3α,7α,12β-Trihydroxy-5β-cholan-24-oic acid (**6**) [25,26]. Method **A**: A solution of a crude mixture of **16** and **18** (250 mg, 0.493 mmol), obtained from a NaBH_4_/CeCl_3_·7H_2_O reduction of **17**, in methanol (10 mL) was treated with an 8 M aqueous solution of potassium hydroxide (1 mL) at 80 °C overnight. Subsequently, methanol was evaporated, and the remainder diluted with water and the pH adjusted to 1 by dropwise addition of a 1 M aqueous solution of hydrochloric acid. The aqueous mixture was extracted with ethyl acetate (3×) and the combined organic fractions were washed with brine, dried over MgSO_4_ and concentrated. The desired reaction product **6** was isolated by automated column chromatography [silica gel (25 µm), acetone (+1% acetic acid)/dichloromethane (+1% acetic acid) 0–60%] using a 12 g high-performance (HP) cartridge, followed by subsequent chromatography on C18-silica gel gradient eluting with water/methanol (0–70%) to yield 78 mg (39%) of **6** as a colorless amorphous solid. 

Method **B**: Employing the same procedure for the preparation of **6** as described in Method A above, a mixture of **20** and **21** (397 mg, 0.806 mmol), which was prepared by a Meerwein–Ponndorf*–*Verley reduction of **17**, was dissolved in methanol (18 mL) and saponified in the presence of potassium hydroxide (8 M; 1 mL) to give 154 mg (47%) of **6** as a colorless amorphous solid (also after additional chromatography on reversed phase silica gel).

${\left[\alpha \right]}_{D}^{20}$ = +32.4 (*c* 0.565, MeOH), [29]: ${\left[\alpha \right]}_{D}^{20}$ = +30.5 (EtOH); ^1^H NMR (500 MHz, CD_3_OD) δ 3.83–3.79 (m, 1H), 3.41–3.33 (m, 2H), 2.34 (ddd, *J* = 15.4, 10.2, 5.3 Hz, 1H), 2.29–2.17 (m, 2H), 2.03–1.87 (m, 4H), 1.84 (dt, *J* = 14.3, 3.0 Hz, 1H), 1.81–1.72 (m, 2H), 1.68–1.59 (m, 3H), 1.59–1.47 (m, 3H), 1.46–1.20 (m, 7H), 1.06–0.96 (overlapping signals: m, 1H and 1.03, d, *J* = 6.9 Hz, 3H), 0.94 (s, 3H), 0.72 (s, 3H); ^13^C NMR (125 MHz, CD_3_OD) δ 178.28, 80.34, 72.81, 68.83, 58.78, 49.90, 48.87 (C from HMBC), 43.01, 40.53, 39.59, 36.63, 36.25, 36.00, 34.06, 33.60, 33.42, 31.96, 31.39, 31.07, 24.83, 24.22, 23.30, 21.65, 8.42; proton and carbon NMR data of **6** were generally consistent with those reported in [25,26]. However, minor inconsistencies were found in the given carbon data. HRMS (ESI) m/z calcd for C_24_H_40_O_5_Na^+^ 431.2768, found 431.2768. 

Mixture of methyl 3α,7α-dihydroxy-14(13→12)*abeo*-5β,12α(H)-chol-13(18)-en-24-oate (**22**) and methyl 3α,7α-dihydroxy-14(13→12)*abeo*-5β-chol-12(13)-en-24-oate (**23**).

Method **A**: To a solution of **18** (603 mg, 1.19 mmol) in dry pyridine (6 mL) was added dropwise trifluoromethanesulfonic anhydride (Tf_2_O; 0.36 mL, 2.14 mmol) at 0 °C. After being stirred at 0 °C for 2 h, the reaction was quenched with water and extracted with ethyl acetate (3×). The organic layers were combined, washed with 1 M aqueous hydrochloric acid solution (1×), water (1×) and brine, and were finally dried over MgSO_4_ and concentrated. The residue was purified by automated column chromatography (silica gel, ethyl acetate/petroleum ether 0–30%) to yield 523 mg (90%) of a mixture of alkenes which contained **22** (35%) and **23** (50%) as the major components along with two minor, unidentified alkene side products A (4%) and B (10%) as determined from the proton NMR.

Method **B**: To a solution of **18** (498 mg, 0.983 mmol) and 4-(dimethylamino)pyridine (DMAP; 740 mg, 6.06 mmol) in dry toluene (60 mL) was added *N*-(5-chloro-2-pyridyl)*bis*(trifluoromethanesulfonimide) (Comins’ reagent; 1.17 g, 2.98 mmol) and the resulting reaction mixture was heated at 130 °C for 3 h. After being cooled to room temperature, the reaction was filtered, and the filtrate concentrated. The residue was purified as described in Method A above to give 404 mg (84%) of a mixture of alkenes as a colorless oil which contained **22** (24%) and **23** (37%) along with three minor, unidentified alkenes A (6%), B (30%) and C (5%) as determined from the proton NMR.

Method **C**: To a solution of **18** (313 mg, 0.618 mmol) in a 1:1 mixture of dry toluene (3 mL) and dry pyridine (3 mL) was added dropwise methanesulfonyl chloride (0.16 mL, 2.07 mmol) at 0 °C. After being stirred for 2 h at the same temperature, the reaction was quenched with water and extracted with ethyl acetate (3×). The combined organic fractions were washed with a 1 M aqueous solution of hydrochloric acid (1×), water (1×) and brine, dried over MgSO_4_ and concentrated to yield 379 mg of crude mesylate as a colorless oil. This was re-dissolved in glacial acidic acid (15 mL) and sodium acetate (719 mg, 8.77 mmol) was added. After heating the reaction at 100 °C for 3 h, acidic acid was evaporated off under reduced pressure. The residue was taken up in water and extracted with ethyl acetate (3×). The extracts were combined, washed with brine, dried over MgSO_4_ and concentrated. Excess acidic acid was removed by repeated co-evaporation with dichloromethane. The residue was purified as described in Method A above to give 198 mg (66%) of a mixture of alkenes which contained **22** (12%) and **23** (31%) along with three minor, unidentified alkenes A (11%), B (45%) and C (2%) as determined from the proton NMR.

3α,7α-Dihydroxy-14(13→12)*abeo*-5β,12α(H)-chol-13(18)-en-24-oic acid (**25**) and 3α,7α-dihydroxy-14(13→12)*abeo*-5β-chol-12(13)-en-24-oic acid (**26**). A solution of a mixture of alkenes **22** and **23** (563 mg, 1.15 mmol), derived from a trifluoromethanesulfonic anhydride-mediated rearrangement reaction (Method A), in methanol (20 mL) was treated with a solution of an 8 M aqueous potassium hydroxide solution (1.5 mL) at 84 °C overnight. After being cooled to room temperature, the solvent was evaporated under reduced pressure and the remainder was diluted with water. Then the pH was adjusted to 1 by introducing a 1 M solution of aqueous hydrochloric acid and the aqueous was extracted with ethyl acetate (3×). The organic fractions were combined, washed with brine, dried over MgSO_4_ and concentrated. The crude mixture of acids was purified by automated column chromatography [silica gel (25 µm), ethyl acetate (+1% acetic acid)/petroleum ether (+1% acetic acid) 0–40%] to give 212 mg (47%) of a 12:1 mixture of **26** and an unidentified isomer as well as 195 mg (43%) of a 10:1 mixture of **25** and **27** (as deduced from the respective ^1^H NMR spectra). The latter was separable by additional automated column chromatography on C18-silica gel gradient eluting with water/methanol 0–80% to afford 117 (26%) of **25** and 14.6 mg (3%) of **27** as a minor side product. Both compounds were obtained as colorless oils.

*3α,7α-Dihydroxy-14(13*→*12)abeo-5β,12α(H)-chol-13(18)-en-24-oic acid* (**25**). R*_f_* = 0.32 (acetone/dichloromethane/acetic acid 3:7:0.1), R*_f_* = 0.44 (C18-silica gel, water/methanol 1:4) ${\left[\alpha \right]}_{D}^{20}$ = +27.9 (*c* 0.61, MeOH); ^1^H NMR (500 MHz, CD_3_OD) δ 4.87 (s, 1H), 4.72 (d, *J* = 1.0 Hz, 1H), 3.97 (dd, *J* = 5.5, 2.8 Hz, 1H), 3.36 (tt, *J* = 11.3, 4.2 Hz, 1H), 2.61 (dt, *J* = 7.2, 10.4 Hz, 1H), 2.38 (ddd, *J* = 15.3, 9.6, 5.5 Hz, 1H), 2.29–2.13 (overlapping signals: m, 3H and 2.18, dt, *J* = 11.6, 13.3 Hz, 1H), 1.99–1.91 (m, 2H), 1.89 (ddd, *J* = 14.8, 5.6, 3.4 Hz, 1H), 1.77–1.61 (m, 6H), 1.61–1.37 (m, 6H), 1.35–1.15 (overlapping signals: m, 3H and 1.19, td, *J* = 14.1, 3.6 Hz, 1H), 0.94 (d, *J* = 6.7 Hz, 3H), 0.89 (s, 3H); ^13^C NMR (125 MHz, CD_3_OD) δ 178.01, 153.05, 109.78, 72.76, 67.94, 50.71, 48.57, 46.91, 42.75, 40.59, 39.08, 37.93, 37.66, 36.94, 36.34, 35.95, 32.84, 31.71, 31.26, 29.96, 27.22 (2C), 22.54, 18.46; HRMS (ESI) m/z calcd for C_24_H_38_O_4_Na^+^ 413.2662, found 413.2678. 

*3α,7α-Dihydroxy-14(13*→*12)abeo-5β-chol-12(13)-en-24-oic acid* (**26**). R*_f_* = 0.38 (acetone/dichloromethane/acetic acid 3:7:0.1); ^1^H NMR (500 MHz, CD_3_OD) δ 4.06–4.01 (m, 1H), 3.37 (tt, *J* = 11.2, 4.2 Hz, 1H), 2.42–2.30 (m, 2H), 2.27–2.09 (m, 4H), 2.00–1.80 (m, 6H), 1.72–1.53 (overlapping signals: m, 5H and 1.60, s, 3H), 1.53–1.39 (m, 3H), 1.27–1.14 (m, 3H), 0.99 (d, *J* = 6.9 Hz, 3H), 0.97–0.88 (m, 1H), 0.85 (s, 3H); ^13^C NMR (125 MHz, CD_3_OD) δ 178.00, 139.64, 126.66, 72.73, 67.51, 51.66, 45.46, 43.24, 43.08, 40.49, 37.80, 36.37, 35.73, 35.71, 35.22, 33.95, 31.77, 29.29, 28.86, 26.38, 23.81, 22.30, 19.17, 17.70; HRMS (ESI) m/z calcd for C_24_H_37_O_4_^−^ 389.2697, found 389.2688. 

*3α,7α-Dihydroxy-14(13*→*12)abeo-5β*,*13α(H)-chol-12(14)-en-24-oic acid* (**27**). R*_f_* = 0.30 (water/methanol 1:4; C18-silica gel); ^1^H NMR (500 MHz, CD_3_OD) δ 4.18 (dd, *J* = 5.4, 2.7 Hz, 1H), 3.35 (tt, *J* = 11.2, 4.2 Hz, 1H), 2.65 (dt, *J* = 7.3, 11.5 Hz, 1H), 2.47–2.41 (m, 1H), 2.39–2.29 (m, 2H), 2.28–2.03 (m, 4H), 1.99–1.89 (m, 3H), 1.83–1.75 (m, 2H), 1.74–1.62 (m, 3H), 1.54 (dt, *J* = 14.8, 1.8 Hz, 1H), 1.49–1.36 (m, 5H), 1.33–1.25 (m, 1H), 1.21 (td, *J* = 13.9, 3.4 Hz, 1H), 0.94 (d, *J* = 5.8 Hz, 3H), 0.92 (s, 3H), 0.83 (d, *J* = 7.0 Hz, 3H); ^13^C NMR (125 MHz, CD_3_OD) δ 178.24, 142.12, 136.15, 72.66, 67.96, 52.11, 44.38, 42.81, 40.85, 39.48, 37.85, 37.04, 35.61, 34.55, 34.02, 33.25, 32.35, 31.86, 30.90, 25.27, 23.60, 22.08, 17.42, 13.31; LRMS (ESI) m/z calcd for C_24_H_37_O_4_^−^ 389.3, found 389.2.

3α,7α-Dihydroxy-14(13→12)*abeo*-5β,12α(H),13β(H)-cholan-24-oic acid (**3**) [15]. To a solution of a mixture of alkenes **22** and **23** (280 mg, 0.573 mmol), obtained from a trifluoromethanesulfonic anhydride-mediated rearrangement reaction (Method A), in ethanol (12 mL) was add 10% palladium on charcoal (62 mg) and the atmosphere exchanged for hydrogen. After complete hydrogenation (as determined by crude LRMS), the reaction was filtered through celite, the celite washed with ethanol and ethyl acetate and the filtrate concentrated and dried under high vacuum. 

The crude hydrogenation product was re-dissolved in methanol (15 mL) and saponified with a solution of an 8 M aqueous potassium hydroxide solution (1.2 mL) at 80 °C overnight. After evaporating the solvent, the residue was re-dissolved in water and the solution acidified to pH 1 by introducing dropwise a 1 M aqueous solution of hydrochloric acid causing a colorless precipitate to form. The aqueous was extracted with ethyl acetate (3×) and the combined organic fractions were washed with brine, dried over MgSO_4_ and concentrated. The crude product was subsequently purified by automated column chromatography [silica gel (25 µm), acetone (+1% acetic acid)/dichloromethane (+1% acetic acid) 0–40%] using a 40 g high-performance (HP) cartridge to give 140 mg of a colorless solid. This was further purified by automated column chromatography on C18-silica gel gradient eluting with water/methanol (0–80%) to yield 83.1 mg (37%) of **3** as a colorless solid. In addition, 11.0 mg (5%) of an unidentified side product were isolated as a colorless solid. The spectroscopic data of **3** were consistent with that reported for **30** in [15].

*N*-(3α,7α-Dihydroxy-14(13→12)*abeo*-5β,12α(H),13β(H)-cholan-24-oyl)taurine sodium salt (**24**). To a solution of **3** (170 mg, 0.433 mmol) in tetrahydrofuran (9 mL) was added dry triethylamine (TEA; 72.4 µL, 0.520 mmol). This was cooled to 0 °C before isobutyl chloroformate (IBCF; 67.4 µL, 0.520 mmol) was introduced dropwise. After being stirred for 1 h at 0 °C, a second portion of dry triethylamine (TEA; 0.12 mL, 0.861 mmol) was added followed by the dropwise addition of a solution of taurine (80 mg, 0.639 mmol) in water (1 mL). The reaction was allowed to warm to room temperature and stirred overnight. Then a 2 M aqueous solution of sodium hydroxide (1.3 mL) was added, and the resulting mixture was concentrated under reduced pressure. The crude product was purified by automated column chromatography on silica gel eluting with methanol/chloroform (0–100%), followed by automated column chromatography on reversed phase silica (C18) gradient eluting with water/methanol (20–100%) to give 119 mg (53%) of **24** as a colorless amorphous solid. ${\left[\alpha \right]}_{D}^{20}$ = –2.99 (*c* 1.0, H_2_O); ^1^H NMR (500 MHz, CD_3_OD) δ 4.21–4.11 (m, 1H), 3.71–3.60 (m, 2H), 3.60–3.50 (m, 1H), 3.17 (t, *J* = 7.1 Hz, 2H), 2.48–2.35 (m, 1H), 2.35–2.18 (m, 2H), 2.16–1.90 (m, 3H), 1.88–1.44 (m, 13H), 1.40–0.85 (m, 15H); ^13^C NMR (125 MHz, D_2_O) δ 175.93, 71.50, 66.49, 50.05, 48.51, 44.38, 44.13, 41.20, 38.61, 38.12, 37.61, 36.49, 35.65, 35.21, 35.05, 34.66, 34.54, 32.88, 30.22, 30.12, 26.18, 25.01, 22.08, 21.76, 17.83, 17.57; HRMS (ESI) m/z calcd for C_26_H_44_NO_6_S^−^ 498.2895, 498.2884

3α-Hydroxy-7-oxo-14(13→12)*abeo*-5β,12α(H),13β(H)-cholan-24-oic acid (**28**) and 3,7-dioxo-14(13→12)*abeo*-5β,12α(H),13β(H)-cholan-24-oic acid (**29**). To a solution of (261 mg, 0.665 mmol) in a 1:1:1 mixture of methanol (3 mL), acetic acid (3 mL) and ethyl acetate (3 mL) was added and potassium bromide (5.2 mg, 0.044 mmol) and the mixture was cooled to 14 °C before bleach (15% aqueous hypochlorite solution; 0.33 mL, 0.802 mmol) was introduced dropwise over a period of 15 min. Progress of the reaction was assessed by TLC analysis and, if necessary, more bleach was added (0.1 mL increments) to drive the reaction to completion. Once the oxidation was complete, solid sodium thiosulfate solution (200 mg) was added. The resulting mixture was diluted with water and extracted with ethyl acetate (3×). The combined organic phases were washed with brine, dried over MgSO_4_ and concentrated. The residue was purified by automated flash column chromatography [silica gel, acetone (+1% acetic acid)/dichloromethane (+1% acetic acid) 2–20%] to afford 158 mg (61%) of the 7-oxo derivative **28** and 36.0 mg (14%) of the 3,7-dioxo side product **29**, both as colorless foams.

*3α-Hydroxy-7-oxo-14(13*→*12)abeo-5β,12α(H*),*13β(H)-cholan-24-oic acid* (**28**). ${\left[\alpha \right]}_{D}^{20}$ = −48.4 (*c* 0.915, CHCl_3_); ^1^H NMR (500 MHz, CD_3_OD) δ 3.50 (tt, *J* = 10.9, 4.3 Hz, 1H), 2.88 (dd, *J* = 13.4, 6.3 Hz, 1H), 2.55 (dd, *J* = 12.5, 11.5 Hz, 1H), 2.36 (ddd, *J* = 15.4, 9.0, 5.5 Hz, 1H), 2.27–2.14 (m, 2H), 2.09–2.01 (m, 1H), 1.99–1.88 (m, 2H), 1.86 (dd, *J* = 13.5, 1.5 Hz, 1H), 1.82–1.61 (m, 6H), 1.61–1.53 (m, 2H), 1.46–1.39 (m, 1H), 1.38–1.27 (m, 2H), 1.26–1.08 (overlapping signals: m, 5H and 1.18, s, 3H), 1.01–0.94 (m, 1H), 0.91 (d, *J* = 6.8 Hz, 3H), 0.86 (d, *J* = 6.3 Hz, 3H); ^13^C NMR (125 MHz, CD_3_OD) δ 214.37, 177.88, 71.41, 54.18, 49.45, 47.57, 47.46, 46.82, 44.97, 39.12, 38.74, 38.61, 35.92, 35.64, 33.51 (2C), 32.54, 30.95, 27.21, 26.23, 22.23, 22.09, 18.50, 17.67; HRMS (ESI) m/z calcd for C_24_H_38_O_4_H^+^ 391.2843, found 391.2844.

*3,7-Dioxo-14(13*→*12)abeo-5β,12α(H),13β(H)-cholan-24-oic acid* (**29**). ${\left[\alpha \right]}_{D}^{20}$ = –41.0 (*c* 1.32, CHCl_3_); ^1^H NMR (500 MHz, CDCl_3_) δ 2.75 (dd, *J* = 13.8, 5.8 Hz, 1H), 2.52–2.41 (m, 2H), 2.40–2.20 (m, 7H), 2.19–2.11 (m, 1H), 2.03–1.94 (m, 3H), 1.82–1.74 (m, 1H), 1.74–1.58 (m, 5H), 1.49–1.41 (m, 1H), 1.29–1.21 (overlapping signals: m, 1H and 1.24, s, 3H), 1.21–1.07 (m, 3H), 1.04–0.97 (m, 1H), 0.91 (d, *J* = 6.8 Hz, 3H), 0.85 (d, *J* = 6.2 Hz, 3H); ^13^C NMR (125 MHz, CDCl_3_) δ 210.43, 210.32, 179.48, 53.46, 47.79, 47.51, 46.02, 45.38, 43.47, 43.29, 37.47, 37.21, 36.96, 36.09, 34.78, 32.44, 32.24, 31.75, 26.01, 24.65, 21.19, 21.06, 17.96, 17.20; HRMS (ESI) m/z calcd for C_24_H_36_O_4_Na^+^ 411.2506, found 411.2512.

3α,7β-Dihydroxy-14(13→12)*abeo*-5β,12α(H),13β(H)-cholan-24-oic acid (**4**). To a solution of **28** (153 mg, 0.392 mmol) in dry *iso*-propanol (8 mL) was added sodium in small pieces at 90 °C (one piece at a time until dissolved) until the sodium did not dissolve anymore and a crust started to form at the surface of the reaction. Another 6 mL of dry *iso*-propanol was added to re-dissolve the formed solid material. This process was repeated one more time before the hot mixture was poured into water and the *iso*-propanol was mostly evaporated (caution: all sodium must be dissolved before the reaction mixture is poured into water!). The remainder was diluted further with water, acidified to pH 1 by adding dropwise a 1–2 M aqueous hydrochloric acid solution and the formed precipitate was extracted with ethyl acetate (3×). The combined organic phases were washed with brine, dried over MgSO_4_ and concentrated. The crude product was purified by automated column chromatography using a high-performance (HP) cartridge [silica gel (25 µm), acetone (+1% acetic acid)/dichloromethane (+1% acetic acid) 0–40%] to yield 68.3 mg of **4** (44%) and 56.5 mg (37%) of **3**, both as colorless foams upon co-evaporation with dichloromethane (3–4×). In addition, 5.8 mg (4%) of a mixed fraction was recovered. ${\left[\alpha \right]}_{D}^{20}$ = +36.9 (*c* 0.425, MeOH); ^1^H NMR (500 MHz, CD_3_OD) δ 3.60–3.53 (m, 1H), 3.50–3.42 (m, 1H), 2.37 (ddd, *J* = 15.1, 9.1, 5.6 Hz, 1H), 2.22–2.14 (m, 1H), 2.01–1.92 (m, 1H), 1.87–1.69 (m, 6H), 1.68–1.44 (m, 9H), 1.44–1.36 (m, 1H), 1.36–1.27 (m, 2H), 1.27–1.12 (m, 3H), 1.03–0.93 (overlapping signals: m, 2H and 0.94, s, 3H), 0.92 (d, *J* = 6.9 Hz, 3H), 0.85 (d, *J* = 6.2 Hz, 3H); ^13^C NMR (125 MHz, CD_3_OD) δ 178.10, 74.14, 72.10, 49.65, 49.53, 46.52, 46.00, 44.31, 43.05, 39.62, 38.45, 38.14, 36.82, 34.92, 33.73 (2C), 31.96, 31.43, 28.38, 26.44, 22.99, 22.30, 18.54, 17.90; HRMS (ESI) m/z calcd for C_24_H_40_O_4_Na^+^ 415.2819, found 415.2821.

### 2.2. Participant Recruitment and Fibroblast Collection

Punch skin biopsies were taken from participants with clinically manifesting sporadic PD. Participants were recruited through Sheffield Teaching Hospitals though the Sheffield Institute of Translational Neuroscience (SITraN). The study was approved by the local institutional review boards (12/YH0367). Written informed consent was obtained from all research participants included in this study. The age of the sPD participants was 53.3 ± 2.5 age in years ± SD. The mitochondrial phenotype has been previously described for the sPD lines [7]. Punch skin biopsies were taken from 3 LRRK2G2019S mutation carriers with clinically manifest PD. The LRRK2G2019S mutant patients included in this study were not directly related to each other. The mean age of the LRRK2G2019S mutant patients at the time of biopsy was 58.6 ± 5.5. The study was approved by the respective local ethics committees. Biopsies were only carried out after informed consent was taken from all research participants. The mitochondrial deficits have been previously published in [30]. 

### 2.3. Fibroblast Culture

Primary fibroblast cells from sPD patients were cultured continually in high glucose (4500 mg/L) Dulbecco’s Modified Eagle’s medium (DMEM) supplemented with 10% fetal bovine serum, 100 IU/mL penicillin, 100 µg/mL streptomycin, 1 mM sodium pyruvate and 50 µg/mL uridine. Primary fibroblast cells from LRRK2 patients were cultured continually in Earles Modified Eagle’s Medium (EMEM) with the same supplements as described above. Unless otherwise stated, 48 h prior to analysis, the glucose-containing media was exchanged for glucose-free DMEM with the same supplementation and in addition, 5 mM galactose. All cells were assessed between passages 6–10.

### 2.4. Measurement of Mitochondrial Function

For mitochondrial membrane potential, cells were plated at a density of 100 cells per well in a black 384-well plate. For the total cellular ATP levels cells were plated at a density of 5000 cells per well of a white-walled 384-well plate. UDCA (Sigma-Aldrich Ltd., St. Louis, MI, USA) or the newly synthesized compounds described herein were added to culture media (0.06, 0.25, 1, 3, 6, 25, 100 and 300 nM) 24 h prior to assay. Dimethyl sulfoxide (DMSO) was used as vehicle control. Total cellular ATP levels were measured using the ATPlite Luminescence Assay System (Perkin Elmer, Waltham, MA, USA) according to the manufacturer’s instruction. All measurements for ATP were subsequently normalized to cell number using the CyQUANT^®^ NF Cell Proliferation Assay kit (Life Technologies, Carlsbad, CA, USA).

### 2.5. Mitochondrial Membrane Potential

Mitochondrial membrane potential was measured using tetramethlyrhodamine (TMRM) staining of live fibroblasts. Briefly; 24 h after drug treatment cells were incubated with 80 nM TMRM and 10 μM Hoechst in phenol red free media for 1 h. Cells were washed and imaged using the InCell Analyzer 2000 high content imager (GE Healthcare, Chicago, IL, USA). Raw images were processed, and parameters obtained using a custom protocol in InCell Developer software (GE Healthcare) allowing for segmentation of mitochondria, nuclei and cell boundaries.

## 3. Results and Discussion

### 3.1. Chemistry

In our previous work, we employed a Nametkin rearrangement to synthesize a new set of 12β-methyl-18-*nor*-bile acids for study [15]. To enable this reaction, the 12-hydroxy group of cholic acid was firstly activated as the mesylate (**8**) (Figure 2A). Upon heating in a protic solvent (acetic acid) in the presence of a mild base (sodium acetate) the 18-methyl group was able to migrate into the equatorial 12β-position, displacing the axial mesylate leaving group. Subsequent elimination of a proton then gave rise to the Δ^13(14)^- and Δ^13(17)^-unsaturated 12β-methyl-18-*nor*-bile acid derivatives **10** and **11**. Based on these considerations, we reasoned that formation of the C-*nor*-D-*homo* framework may be favored if the leaving group at position 12 was situated on the β-face, thus promoting migration of the C13-C14 bond, whilst limiting movement of the 18-methyl group (Figure 2B). This notion was supported by a literature survey which revealed that such a rearrangement was reported to occur in the biosynthesis of cyclopamine (**5**) and other C-*nor*-D-*homo* veratrum alkaloids [31,32,33,34,35,36]. Moreover, this concept was implemented by Giannis and co-workers in the total synthesis of cyclopamine (**5**) [17,37,38].

To start our synthetic efforts, we aimed to prepare the diacetate-protected lagocholic acid methyl ester **18** as the key intermediate (Figure 1). Therefore, we oxidized the protected cholic acid precursor **16** to the corresponding 12-ketone **17** which was subsequently reduced with borane *tert*-butylamine complex to yield a 1:2 mixture of the 12α- and 12β-alcohols **16** and **18**, respectively [25,26,27]. With the availability of suitable crystals, the structure of **18** was confirmed by X-ray crystallography. Attempts to perform the same reaction under Luche (Figure 1, Method B) or Meerwein–Ponndorf–Verley (MPV) reaction conditions (Figure 1, reaction **17**→**20**/**21**), however, led to a less favorable reaction outcome [39,40]. Although the desired 12β-alcohol **18** was formed using Luche conditions, the reaction proved less selective with an overall product ratio of 1:1.4. Application of the MPV reaction conditions favored reduction of the 12-oxo-function to the desired 12β-alcohol in a good ratio of 1:7, but the harsh reaction conditions caused deacetylation of the 3-hydroxy group and transesterification to the corresponding *iso*-propyl esters **20** and **21**. To assign the reaction products **20** and **21** from the MPV reaction, the mixture of isopropyl esters was saponified, confirming lagocholic acid **6** as the major product and cholic acid as the minor component of the mixture.

Subsequent treatment of **18** with triflic anhydride (Tf_2_O) in pyridine at 0 °C induced the desired rearrangement affording an inseparable mixture of alkenes (Figure 2) [41,42]. Compounds **22** and **23** were the two major reaction products, present in a 1.3:1 ratio. When the reaction was conducted with Comins’ reagent, or a mesylate-activated starting material, the reaction proceeded less cleanly generating **22** and **23** in diminished yields [17,43]. Basic hydrolysis of the mixture of alkenes **22** and **23** allowed chromatographic separation and characterization of the corresponding deprotected alkene acids **25** and **26**. Additional purification of **25** on reversed-phase silica gel led to the recovery of a minor isomer, the putative structure of which was assigned to be **27** by extensive 2D-NMR analysis. Hydrogenation of a mixture of alkenes **22** and **23** derived from the triflic anhydride-mediated rearrangement, followed by hydrolysis furnished the 14(13→12)-*abeo*-bile acid **3**. The spectroscopic data of **3** were identical to those of the same compound (**3**) isolated as a minor side product from the Nametkin rearrangement (Figure 2A) as reported previously [15]. Taurine conjugate **24** was prepared by the reaction of taurine with the mixed acyl carbonate formed from carboxylic acid **3** and isobutylchloroformate (IBCF) [44].

To prepare the 7β-configured analogue **4**, compound **3** was subjected to a bleach/KBr-mediated oxidation which formed the corresponding 7-ketone (**28**) selectively in 61% yield (Figure 3) [45]. In addition, overoxidation produced a small amount of the diketone analogue **29** as a side product. [46]. Finally, reduction of **28** with sodium in *iso*-propanol yielded a mixture of **4** and **3**, from which compound **4** was separated by column chromatography and obtained in 44% yield. 

### 3.2. Biology

Compounds **3**, **4**, **7**, **24**, **25** and **29** were initially screened for potential efficacy on mitochondrial membrane potential and cellular ATP levels in fibroblasts from patients with sporadic PD or mutations in LRRK2 at a single concentration, 100 nM (Appendix A). From this initial evaluation, compounds **3**, **4**, **7** and **24** emerged as screening hits that were found to increase the mitochondrial membrane potential and cellular ATP levels above the vehicle control whereas compounds **25** and **29** were inactive. Therefore, **3**, **4**, **7** and **24** were investigated further on fibroblasts from three sPD patients and three LRRK2 mutant patients at an eight-point concentration-response curve to test their effect on increasing the total cellular ATP and mitochondrial membrane potential. No biological data was generated for compounds **26** and **27** as these proved to be not stable and degraded.

All compounds were tested against UDCA for comparison in the same PD patient fibroblast lines in the same experiments. Compound **3** increased cellular ATP levels to a maximum of 278% (where DMSO-treated sPD fibroblasts are set to 100%), however, as can be deduced from Figure 3 the effect was variable between the sPD fibroblast lines tested. Indeed, only one data point is included from one sPD fibroblast line at the highest concentration as this concentration (300 nM) proved toxic to two sPD fibroblast lines. The maximal improvement is greater than UDCA which reaches a maximum of 165% increase. Compounds **4**, **7** and **24** all increased cellular ATP levels to a similar extent, varying between 139–150%. Noticeably, the response was far more consistent across all three sPD fibroblast lines after treatment with **4**, **7** and **24** indicating a robust and uniform response to compound treatment. The maximal response varies between compounds; the EC_50_ also varies between the compounds with **3** having a higher EC_50_ than UDCA, and **4**, **7** and **24** having lower EC_50_ values than UDCA (Table 1).

To assess compound effect in a genetically more homogeneous population, we also assessed total cellular ATP levels in fibroblasts from patients with LRRK2G2019S mutation, in which we have previously shown a restoration with UDCA treatment [47]. In these patient-derived fibroblasts, the compounds produced a higher maximal recovery than UDCA (ranging between 140–173% compared to 127% by UDCA), in addition lower EC_50_; hence they are more efficacious than UDCA in LRRK2G2019S mutant fibroblasts. Table 1 shows the % maximal recovery and EC_50_ for each compound and the concentration response curves are shown in Figure 4 compared to UDCA.

We also investigated the effect of the compounds (**3**, **4**, **7** and **24**) on mitochondrial membrane potential again in comparison to UDCA. Contrary to cellular ATP levels, compound **3** did not show any effect at all on mitochondrial membrane potential levels in any sPD fibroblast lines (Figure 5). In contrast, UDCA produced a maximal effect of 125% of the DMSO vehicle treated levels which was very similar to the maximal effect produced by treatment with **4** and **7**. However, although producing a similar maximal increase, this increase after treatment with **4** and **7** was only evident at higher concentrations, therefore the EC_50_ is higher for both of those compounds in comparison to UDCA (Table 2). Finally, compound **24** did not produce a robust increase in mitochondrial membrane potential across the three sPD fibroblast lines. The effect of the compounds in the genetically more homogenous LRRK2G2019S fibroblasts contrasted with the sPD fibroblast lines. Compound **24** was also not active in these lines; however, compound **4** was very active in the LRRK2 fibroblast lines, with higher maximal effect and lower EC_50_ concentration than UDCA. Compounds **4** and **7** were also both active in the LRRK2 mutant fibroblast lines as shown in Table 2 and Figure 6.

## 4. Conclusions

Here we have described the synthesis of a series of C-*nor*-D-*homo* bile acid analogues and the methylated lagocholic acid derivative **7** which, upon treatment of sPD and LRRK2G2019S fibroblasts, show an improvement in the mitochondrial phenotype of these patient-derived cell lines. Interestingly, we are able to discern differences between different metabolic measures between compounds. For example, treatment with compound **3** elicits a large maximal effect on cellular ATP levels, to a greater extent than UDCA, but no effect on mitochondrial membrane potential in the sporadic PD fibroblast lines. Conversely, compounds **4** and **7** have similar maximal effects to UDCA on both cellular ATP and mitochondrial membrane potential; however, they are more potent than UDCA with lower EC_50_ values. Of particular interest is that all the novel bile acid analogues were more efficacious than UDCA when tested in the more homogeneous LRRK2G2019S patient fibroblasts. Our previous work found that UDCA restores mitochondrial function in multiple PD patient fibroblast lines, including from those with mutations in *parkin* or LRRK2 and sporadic PD [7,14,48]. However, a lower concentration was needed to achieve maximal recovery in the LRRK2 mutant fibroblast lines [47]. This fits with the data presented here, perhaps indicating that there is a mechanistic interaction of bile acids with LRRK2 or the bile acids interact specifically with the cellular pathology caused by LRRK2. This indicates that the effect on mitochondrial membrane potential and cellular ATP can be separated from each other, perhaps, as they are fundamentally due to differing mechanisms. The exact cellular mechanism by which UDCA elicits these effects is not clear. Although modulation of the Akt signaling pathways and glucocorticoid receptor activation [49,50,51] has been reported before, the exact cellular target is not known. These data may suggest that multiple mechanisms are involved which could be investigated and fine-tuned by further structural manipulation of the compounds presented here. Additional research will be required in order to investigate the cellular target of these novel bile acid analogues, to understand their mode of action and indeed if there is a mechanistic link with LRRK2 biology.

Several studies have also shown that the profile of bile acids and their metabolites is altered in patients with PD [52,53]. This is an important consideration if progressing any bile acid therapeutic to the clinic for PD. Indeed, with UDCA in clinical trials for PD, the potential neuroprotective effect of bile acids and their conjugates is also an area of active research for other neurodegenerative diseases. Altered bile acid profiles have also been found in Alzheimer’s Disease and active clinical trials with TUDCA are underway in Alzheimer’s and Motor Neuron Disease, in addition to large quantities of preclinical data showing protective effects of bile acids in models for these conditions. Indeed, the recent FDA approval of a TUDCA-containing mixture from Amylyx highlights the potential of this area of research for clinical benefit. This study adds to the landscape of new bile acid chemistry which could have a potentially protective or restorative effect across multiple neurodegenerative diseases and provides further scaffolds to understand the mechanism by which these mitochondrial effects are mediated. Further work needs to be carried out to determine the mechanism of action and in vivo efficacy of these compounds.

## Data Availability

All data are reported in this publication and the Appendix A.

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
