# Peer review of "3α,7-Dihydroxy-14(13→12)abeo-5β,12α(H),13β(H)-cholan-24-oic Acids Display Neuroprotective Properties in Common Forms of Parkinson’s Disease"

_biomolecules, 2022, doi:10.3390/biom13010076_

Round 1
Reviewer 1 Report
The manuscript entitled "3α,7-Dihydroxy-14(13→12)abeo-13-epi-5β-cholan-24-oic Acids 2 Display Neuroprotective Properties in Common Forms of Par-3 kinson's Disease" submitted by Luxenburger et all shows the synthesis of new C-nor-D-homo bile acids and their biological study in Parkinson's disease. In general, this manuscript covers a topic of great interest and presents interesting biological results. I would recommend its publication if the following points are resolved:
1- In the study of biological activities only compounds 3, 4, 7 and 24 were evaluated, I wonder why only those compounds? Wouldn't it be interesting to know what biological properties the other intermediates have? A solid rationale for why only these compounds were studied should be included in the article text. But, if they were actually studied and not included preliminarily, I suggest including them in the study.
2- In section 3.1 of the manuscript (Results and Discussion, Chemistry) the discussion for obtaining compound 4 (scheme 3) is missing, it must be included in the manuscript.
3- In section 3.1, mention Diagram 2 in the text (between lines 544 to 558).
4- In section 1, mention Figure 1 in the text and change the position of Figure 1 to line 89-90.
5- In section 2.1.2 it is necessary to include the characterization data of compound 3.
6- After the conclusion, in lines 704 to 709, it is mentioned "The Supporting Information is available free of charge on the ACS Publications website", it must be updated to the Biomolecules format.
7- Supplementary Materials. Currently there are only the biological evaluations and schemes 1, 2 and 3, in addition the reagents of each reaction must be indicated as a footnote to the scheme.
8- Supplementary Materials. Copies of all 1D and 2D NMR characterizations mentioned in the manuscript must be included. A copy of the HRMS for each compound must also be included.
9- A purity analysis of the compounds that were biologically evaluated must be provided. This should be included in both section 2.1.2 and its respective copy in the Supplementary Materials.
Suggestion:
For future syntheses, I recommend evaluating Noyori's Asymmetric Transfer Hydrogenation (try both enanteomers, RuCl(p-cymene)[(S,S)-Ts-DPEN] and RuCl(p-cymene)[(R,R)-Ts -DPEN] ) as an alternative to the use of Luche and above all to the use of the Meerwein-Ponndorf-Verley reaction due to the use of mercury salts. Another alternative is the reduction Corey CBS
Author Response
Reviewer 1.
The manuscript entitled "3α,7-Dihydroxy-14(13→12)abeo-13-epi-5β-cholan-24-oic Acids 2 Display Neuroprotective Properties in Common Forms of Parkinson's Disease" submitted by Luxenburger et all shows the synthesis of new C-nor-D-homo bile acids and their biological study in Parkinson's disease. In general, this manuscript covers a topic of great interest and presents interesting biological results. I would recommend its publication if the following points are resolved:
- In the study of biological activities only compounds 3, 4, 7 and 24 were evaluated, I wonder why only those compounds? Wouldn't it be interesting to know what biological properties the other intermediates have? A solid rationale for why only these compounds were studied should be included in the article text. But, if they were actually studied and not included preliminarily, I suggest including them in the study.
We thank the reviewer for pointing this out. Some of the intermediaries were indeed studied (the ones which were available in large enough quantities for biological measurements). These included 25 and 29. However, these compounds showed no activity in the primary assays across sPD and LRRK2 patient lines when tested at 2 concentrations (data table added to supplementary data: Table S1). Therefore, these were not taken on for further evaluation in the following, more advanced assay work that we discuss in this paper. In addition, compounds 26 and 27 proved to be not stable and degraded. To better clarify this, we have added text into the manuscript.
2- In section 3.1 of the manuscript (Results and Discussion, Chemistry) the discussion for obtaining compound 4 (scheme 3) is missing, it must be included in the manuscript.
We sincerely thank the reviewers for highlighting this. This has now been added.
3- In section 3.1, mention Diagram 2 in the text (between lines 544 to 558).
We have added “Scheme 2” as suggested.
4- In section 1, mention Figure 1 in the text and change the position of Figure 1 to line 89-90.
We have moved Figure 1 and added “Figure 1” to the text as suggested.
5- In section 2.1.2 it is necessary to include the characterization data of compound 3.
The experimental for compound 3 was actually included between lines 362-381. However, in the submitted manuscript the compound was wrongly numbered as “4”. We have therefore corrected this to “3” in lines 362, 379, 381 and 383. We apologies if this has caused confusion.
In regard to the characterization data, we say in line 381 that “The spectroscopic data of 3 was consistent with that reported for 30 in [15].” In our view this should be sufficient since we have reported the isolation and full characterization (including structural determination by X-ray) in our previous paper: ACS Omega 2021, 6, 25019-25039, 762 doi:10.1021/acsomega.1c04199.
However, for convenience we have included a 1H and 13C NMR spectrum and, as requested, an HRMS plot in the supporting information.
6- After the conclusion, in lines 704 to 709, it is mentioned "The Supporting Information is available free of charge on the ACS Publications website", it must be updated to the Biomolecules format.
We have amended the section according to reflect the standards of Biomolecules. However, a link to the actual supporting information will have to be added by the journal.
7- Supplementary Materials. Currently there are only the biological evaluations and schemes 1, 2 and 3, in addition the reagents of each reaction must be indicated as a footnote to the scheme.
8- Supplementary Materials. Copies of all 1D and 2D NMR characterizations mentioned in the manuscript must be included. A copy of the HRMS for each compound must also be included.
9- A purity analysis of the compounds that were biologically evaluated must be provided. This should be included in both section 2.1.2 and its respective copy in the Supplementary Materials.
Regarding point 7 - 9:
We actually had prepared a supporting information file for this submission, but unfortunately it was not attached to this submission. It may not have uploaded properly. Our apologies.
We have now made sure this information is included. It should address the reviewer’s comments.
The supporting information file includes the corresponding HPLC traces regarding assessment of ratios and purity, 1H and 13C NMR spectra and as requested HRMS spectra, X-ray data for compound 18 and 2D NMR spectra for 27.
For future syntheses, I recommend evaluating Noyori's Asymmetric Transfer Hydrogenation (try both enanteomers, RuCl(p-cymene)[(S,S)-Ts-DPEN] and RuCl(p-cymene)[(R,R)-Ts -DPEN] ) as an alternative to the use of Luche and above all to the use of the Meerwein-Ponndorf-Verley reaction due to the use of mercury salts. Another alternative is the reduction Corey CBS
We sincerely thank the reviewer for these suggestions. We will definitely look into these options to support our ongoing research program.
Reviewer 2 Report
The submission presents data obtained for novel C-nor-D-homo bile acid derivatives and the 12-hydroxy-methylated derivative of lagocholic acid in fibroblasts from patients with either sporadic or LRRK2 mutant PD. The topic sounds both interesting and relevant, especially because ursodeoxycholic acid (UDCA) has been identified previously as a bile acid which leads to increased mitochondrial function in multiple in vitro and in vivo models of PD. Overall, the manuscript is written understandable. The synthesis of the compounds and all the steps are described. However, in my opinion, there are some major points or questions that need to be considered for publication.
1. The Authors have not followed the preparation of the „supplementary materials” according to the journal’s instructions.
2. Minor typographical errors were found throughout the manuscript and should be amended.
-Lines 28-30 – Could the Authors change the sentence: „These compounds perform at least as well as UDCA (…)” it is hard to follow;
-lines 22, 49– Could the Authors consider adding the hyphen in „disease modifying”.
- Consider adding an article in „synthesis”
- prevalence - It seems that there is a pronoun problem here, I suggest adding „its”
- Line 52- Consider adding a comma.
- Line 64- It seems that there is an article usage problem in „mitochondrial”
- Line 76- a missing dot in a sentence
- Line 80- Consider using a plural demonstrative or a singular noun instead. It should be „that bile acid” or „those bile acids”.
- Line 90- Consider using a plural demonstrative or a singular noun instead.
- Line 92- Consider changing the verb form from „was” to „were”
- Line 110- NMR spectra without a hyphen
- Line 119- Should be „a calibrant” and „high-resolution”
- Line 150-152- „To a solution of (…)” Could the authors check the correctness of this sentence, it is difficult to understand, moreover „over the period” can be changed into „for”
- Line 195 „reaction” is repeted
- 217 MgSO4- „4” in subscript
- Line 235- unify the notation (x)
- Line 457- it should be „through”
- Line 458 approval number of the local review boards
- Line 460- Please change this sentence: „The age of the participants was age in (…)”
- Line 478-it should be „passages”
-Line 546- it should be „reaction”
-Line 557- please add „the” before „reaction”
- Line 572- Consider adding the hyphen in „concentration response”
- Line 577-it should be „data point”
- Line 582- it should be „indicating”
- Line 584- the sentence „As can be seen from Table 1 not only did the maximal response vary between compounds (…) should be revised.
- Line 586- it should be „the compound”
- Line 675- the sentence „Our previous work using UDCA showed UDCA (…) should be revised.
- UDCA is repeated;
- it will be hard for readers to understand the meaning of the word „parkin”
- Line 697- It seems that „TUDCA containing” is missing a hyphen.
-Line 698- It seems that preposition use may be incorrect here
3. In my opinion, all the numbers in the molecular formula should be written using subscripts; lines: 112, 155, 165, 189, 190, 208, 226 and many more, please check the manuscript carefully. Consider also lines: 124 (F254), and 87 (IC50). The same situation in the case of the superscript (line 190 and many more, check carefully) positions of „+”, it should be C29H46O7Na+ and „-„ in C24H38O4-)
4. There are no conclusions in this study (except the effect on mitochondrial membrane potential and cellular ATP), only hypothesis (mechanisms of action) and suggestion (further structural manipulation). Whether the Authors intend to study the selected structures in other models of PD (in vitro or in vivo) to confirm their activity?
Author Response
Reviewer 2.
The submission presents data obtained for novel C-nor-D-homo bile acid derivatives and the 12-hydroxy-methylated derivative of lagocholic acid in fibroblasts from patients with either sporadic or LRRK2 mutant PD. The topic sounds both interesting and relevant, especially because ursodeoxycholic acid (UDCA) has been identified previously as a bile acid which leads to increased mitochondrial function in multiple in vitro and in vivo models of PD. Overall, the manuscript is written understandable. The synthesis of the compounds and all the steps are described. However, in my opinion, there are some major points or questions that need to be considered for publication.
- The Authors have not followed the preparation of the „supplementary materials” according to the journal’s instructions.
As mentioned before, we actually had prepared a supporting information file for this submission, but it may not have uploaded properly. Our apologies. We have now made sure this information is included.
- Minor typographical errors were found throughout the manuscript and should be amended.
We really would like to thank the reviewer for highlighting these. They are of great value.
-Lines 28-30 – Could the Authors change the sentence: „These compounds perform at least as well as UDCA (…)” it is hard to follow;
We have changed the text to:
“These compounds boost mitochondrial function to a similar level or above that of UDCA in many assays”
-lines 22, 49– Could the Authors consider adding the hyphen in „disease modifying”.
Done
- Consider adding an article in „synthesis”
We assume the reviewer thought of line 26 here:
“..we describe synthesis and biological evaluation of novel…”.
We have adopted the reviewer’s suggestion and added a “the”.
- prevalence - It seems that there is a pronoun problem here, I suggest adding „its”
Added
- Line 52- Consider adding a comma.
Added
- Line 64- It seems that there is an article usage problem in „mitochondrial”
We have deleted the article.
- Line 76- a missing dot in a sentence
Added
- Line 80- Consider using a plural demonstrative or a singular noun instead. It should be „that bile acid” or „those bile acids”.
We are not sure what the reviewer had in mind here.
We referred to the unique C/D-ring system and then we say “..that bile acids of this type [featuring this particular ring set up]…are not known..” Because we believe that there is always more than one we prefer to keep as is.
- Line 90- Consider using a plural demonstrative or a singular noun instead.
We have changed this sentence to:
“In this paper we describe the synthesis of 3, its taurine conjugate (24) as well as the 7β-configured analogue 4”
- Line 92- Consider changing the verb form from „was” to „were”
Changed
- Line 110- NMR spectra without a hyphen
Changed
- Line 119- Should be „a calibrant” and „high-resolution”
Added
- Line 150-152- „To a solution of (…)” Could the authors check the correctness of this sentence, it is difficult to understand, moreover „over the period” can be changed into „for”
We have changed to: “To a solution of 16 (6.76 g, 13.3 mmol) and sodium acetate (13.6 g, 166 mmol) in methanol (100 mL) was added bromine (6.2 mL, 121 mmol) dropwise over 3 h at room temperature.”
- Line 195 „reaction” is repeted
We replaced “After complete reaction (TLC analysis)” with “Then”. It reads now: “Then the reaction was quenched with water and extracted with ethyl acetate (3×).”
- 217 MgSO4- „4” in subscript
Changed
- Line 235- unify the notation (x)
Changed, and in line 215
- Line 457- it should be „through”
Changed
- Line 458 approval number of the local review boards
Added
- Line 460- Please change this sentence: „The age of the participants was age in (…)”
Changed
- Line 478-it should be „passages”
Changed
-Line 546- it should be „reaction”
Changed
-Line 557- please add „the” before „reaction”
Changed
- Line 572- Consider adding the hyphen in „concentration response”
Added
- Line 577-it should be „data point”
Changed
- Line 582- it should be „indicating”
Changed
- Line 584- the sentence „As can be seen from Table 1 not only did the maximal response vary between compounds (…) should be revised.
Altered to “The maximal response varies between compounds, the EC50 also varies between the compounds with 3 having a higher EC50 than UDCA, and 4, 7 and 24 having lower EC50 values than UDCA (Table 1).”
- Line 586- it should be „the compound”
Altered as above
- Line 675- the sentence „Our previous work using UDCA showed UDCA (…) should be revised.
- UDCA is repeated;
- it will be hard for readers to understand the meaning of the word „parkin”
Altered to “Our previous work found that UDCA restores mitochondrial function in multiple PD patient fibroblast lines, including from those with mutations in parkin or LRRK2 and sporadic PD [7,14,45] however, a lower concentration was needed to achieve maximal recovery in the LRRK2 mutant fibroblast lines [44].”
- Line 697- It seems that „TUDCA containing” is missing a hyphen.
Added
-Line 698- It seems that preposition use may be incorrect here
Altered to “Indeed, the recent FDA approval of a TUDCA-containing mixture from Amylyx highlights the potential of this area of research for clinical benefit.”
In my opinion, all the numbers in the molecular formula should be written using subscripts; lines: 112, 155, 165, 189, 190, 208, 226 and many more, please check the manuscript carefully. Consider also lines: 124 (F254), and 87 (IC50). The same situation in the case of the superscript (line 190 and many more, check carefully) positions of „+”, it should be C29H46O7Na+ and „-„ in C24H38O4-)
All are subscript and superscript. This is just the nature of the font “Palatino Linotype” (as per the journal’s guidelines/template).
- There are no conclusions in this study (except the effect on mitochondrial membrane potential and cellular ATP), only hypothesis (mechanisms of action) and suggestion (further structural manipulation). Whether the Authors intend to study the selected structures in other models of PD (in vitro or in vivo) to confirm their activity?
Added “Further work needs to be carried out to determine the mechanism of action and in vivo efficacy of these compounds.”